# Learning Granular Media Avalanche Behavior for Indirectly Manipulating Obstacles on a Granular Slope

**Haodi Hu**, **Feifei Qian**[†], **Daniel Seita**[†]
[†]Equal advising
University of Southern California, United States
`haodihu@usc.edu`

**Abstract:** Legged robot locomotion on sand slopes is challenging due to the complex dynamics of granular media and how the lack of solid surfaces can hinder locomotion. A promising strategy, inspired by ghost crabs and other organisms in nature, is to strategically interact with rocks, debris, and other obstacles to facilitate movement. To provide legged robots with this ability, we present a novel approach that leverages avalanche dynamics to indirectly manipulate objects on a granular slope. We use a Vision Transformer (ViT) to process image representations of granular dynamics and robot excavation actions. The ViT predicts object movement, which we use to determine which leg excavation action to execute. We collect training data from 100 real physical trials and, at test time, deploy our trained model in novel settings. Experimental results suggest that our model can accurately predict object movements and achieve a success rate $\geq 80\%$ in a variety of manipulation tasks with up to four obstacles, and can also generalize to objects with different physics properties. To our knowledge, this is the first paper to leverage granular media avalanche dynamics to indirectly manipulate objects on granular slopes. Supplementary material is available at https://sites.google.com/view/grain-corl2024/home.

**Keywords:** Granular media, Avalanche dynamics, Legged robots, Manipulation of deformable substrate.

## 1 Introduction

Legged locomotion across granular surfaces such as sand is a formidable challenge due to factors such as insufficient support offered by the sand surface and the complex dynamics of leg-sand interactions [1, 2, 3]. It is particularly challenging for robots to climb up steep granular slopes, as the sand could easily flow underneath robot legs due to the reduced shear resistance forces [4]. Recent studies on obstacle-aided robot locomotion show the potential for legged robots to strategically leverage large obstacles within sand, such as rocks and boulders, to traverse granular and uneven terrains [5, 6, 7, 8]. However, such "obstacle-aided locomotion" strategies require specific leg-obstacle contact locations [5], thus the ability to move rocks and boulders to desired locations became essential for these strategies to apply. To address this challenge, this study aims to propose a method for a legged robot to effectively reposition obstacles on granular slopes by primarily leveraging indirect manipulation. Prior work has shown that external disturbances on a sand incline can trigger avalanche behavior [9, 10, 11], suggesting the potential for a legged robot to exploit this property to advantageously relocate obstacles on a granular surface.

In this work, we propose **G**ranular **R**obotic **A**valanche **IN**teraction (GRAIN), a novel learning-based method for leveraging granular avalanche dynamics for indirectly manipulating objects on a granular slope. Due to a lack of accurate simulators for simulating legged robots and avalanche behavior on granular slopes, we do all experiments in the real world. We use an RHex [12] family robot leg as an external disturbance source which performs excavation actions within a grain tank with mechanical

8th Conference on Robot Learning (CoRL 2024), Munich, Germany.

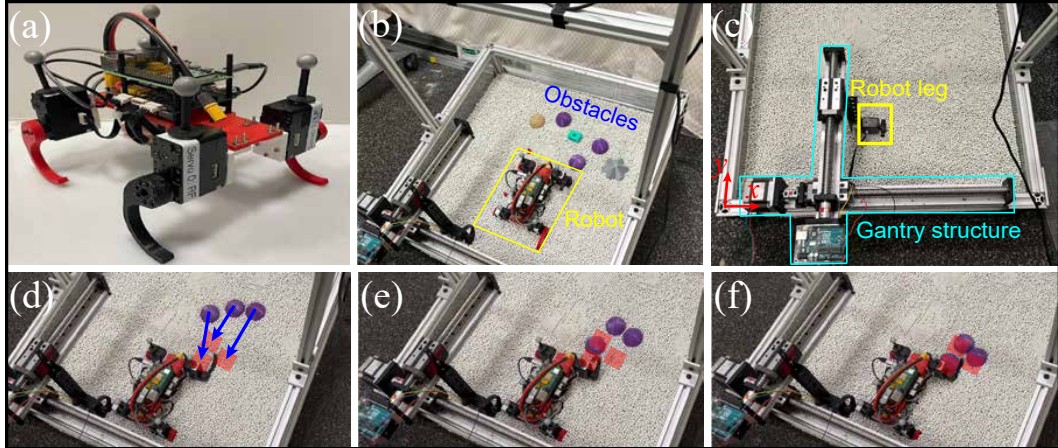

Figure 1: **Top**: (a) our quadrupedal robot and (b) a granular slope with the robot and obstacles; (c) our setup to collect data from one robot leg (highlighted in yellow) manipulating obstacles, with our designed gantry system (highlighted in cyan) for moving the leg over the granular slope. **Bottom**: (d), (e), (f) an example trial of our proposed system to manipulate obstacles on the granular slope. The three blue arrows in (d) represent the change in each obstacle's location after applying the excavation, and the red shaded areas are targets.

support to form a slope. We also design a gantry structure to enable the robot leg to move to different positions for performing leg excavations. We collect training data through physical interaction and deploy our trained model in the real world with a quadrupedal robot. See Figure 1 for our setup.

Accurately predicting the granular media dynamics is essential for our proposed task, but is fundamentally challenging [1, 11, 13, 14]. To address this, we leverage learning methods in this study. We place a rigid 3D-printed obstacle on the granular slope and collect data by applying continuous robot leg excavation actions to investigate avalanche behavior. We represent the granular surface state via the current depth image and the change in depth between sequential robot leg excavations, and we also use an image to represent the leg excavation action. This enables a unified image-based input to our granular media dynamics model. We train a ViT [15] to take our proposed granular media and robot excavation action representations as input and output the object movement on the granular slope. Experiments over 30 trials suggest that our model can accurately predict object movements on a granular slope, and can generalize to objects with different physics properties.

In summary, our main contributions are as follows:

1. A novel problem formulation of using legged robots and leg excavation actions to indirectly manipulate obstacles to targets over complex, granular terrains.
2. GRAIN, a novel approach that uses image-based representations of granular dynamics and legged excavation actions to predict object movements on a granular surface.
3. Experimental results showing that a legged robot using GRAIN can move obstacles to targets, and that GRAIN outperforms an ablation and a non-learning baseline.

## 2 Related Work

**Robot interaction with granular media:** Researchers have explored robotics and granular media in locomotion and manipulation contexts. For example, researchers have enabled crawling [16, 4], hopping [17, 18], running [19, 2, 3], climbing [20], and rapid turning [21] over granular media by leveraging advances in granular force models [1, 13] and machine learning methods [22, 23]. We focus on the orthogonal task of leveraging the properties of granular media to manipulate objects on it. This requires some understanding of how granular media behave, which could be from direct physics analysis [24, 25, 26] or learned models [27, 28]. These models can aid common manipulation tasks involving granular media, which include pouring [29, 30, 31, 32, 33, 34], scooping and bulldozing [35], trenching [36, 14] or adjusting soil with plates [37]. Among relevant prior

work, Wang et al. [38] and Xue et al. [39] study particle-based and density field-based representations, respectively, to learn dynamics models for planar tabletop manipulation of granular media, and Schenck et al. [40] use image representations and train a convolutional neural network to predict changes in the granular media from scooping or pouring actions. As in [40], we use image-based representations of granular media, but our actions are based on leg excavations instead of scooping, pouring, or other tabletop manipulation actions. Moreover, we use legs of a legged robot to indirectly manipulate objects on granular media surfaces.

**Locomotion over diverse terrains:** Recent research has proposed methods that enable legged robots to traverse over a variety of terrains, including outdoor settings that may have sand, vegetation, rocks, or other granular media. A promising technique is reinforcement learning coupled with advanced simulators, which can help avoid significant manual engineering [41, 42]. To better handle diverse terrains, one line of work proposes *adaptation* methods, either via rapid motor adaption [43, 44] or by encoding a family of gait methods which facilities tuning to new terrains [45]. Other works study navigation over challenging terrains [46], which may involve climbing, jumping [47], and crawling under parkour-style settings [48, 49]. While impressive, these works study locomotion over terrains that are much sturdier than our granular surface. Furthermore, they primarily consider locomotion, whereas we focus on manipulating objects on a granular surface.

**Manipulation with legged robots:** While legged robots primarily use legs to move to a target location, they can also use legs for *manipulation*. For example, researchers have proposed methods for using legs to kick soccer balls [50, 51] and to push obstacles [52, 53]. A legged robot can also use two legs to stand up to better enable other legs to press against higher objects such as door buttons [54]. Other works mount an arm on top of a legged robot, and leverage methods such as optimization [55, 56] or machine learning [57, 58, 59] to allow the arm to manipulate objects. These works use legged robots for manipulation via direct contact with an object. To our knowledge, our work is the first to show a legged robot *indirectly* manipulating an object to make it reach a desired pose. To do this, the robot adjusts a granular surface that supports the object.

## 3   Preliminaries and Problem Statement

We consider the RHex [12] family of legged robots, with 1 DOF for each leg. We generate an *excavation action*, per leg, by commanding the leg to rotate at a constant angular speed for one circular cycle. We assume the robot lies on a granular (sand-like) surface which has a slope of $\Phi$ degrees. This surface has $K \geq 1$ rigid obstacles, and we indicate their respective positions at a given time $t$ as $\{\mathbf{s}_t^{(1)}, \ldots, \mathbf{s}_t^{(K)}\}$. The task is to move all obstacles from their initial locations to pre-specified desired locations $\{\mathbf{p}^{(1)}, \ldots, \mathbf{p}^{(K)}\}$. After doing this, we may also want to move the obstacles to a second target location, and we indicate these optional (per-obstacle) target locations as $\{\mathbf{q}^{(1)}, \ldots, \mathbf{q}^{(K)}\}$. Here, each $\mathbf{s}_t^{(k)} \in \mathbb{R}^2$, $\mathbf{p}^{(k)} \in \mathbb{R}^2$, and $\mathbf{q}^{(k)} \in \mathbb{R}^2$ for $k \in \{1, 2, \ldots, K\}$, since we specify 2D positions over an image of the granular surface. For notational convenience, when $K = 1$ we may suppress the superscript $(k)$. We assume access to an overhead camera, which provides $(H \times W)$ *depth* image observations $\mathbf{x}_t$. The objective is to learn a policy which produces a leg excavation action $\mathbf{a}_t$ at time $t$, where the robot rotates one of its legs. A *trial* consists of executing the robot's policy until a termination criteria. We evaluate a trial's performance by averaging the mean absolute error (MAE) distance among all obstacle positions and their respective target positions. We also use MAE to evaluate models which predict where obstacles move based on robot actions.

## 4   Approach: GRAIN

### 4.1   Image Representations of Granular Dynamics

Prior work has shown that external disturbances can trigger avalanche behavior on granular slopes [60]. Inspired by this, we aim to leverage robot leg excavation actions to cause avalanche behaviors to move obstacles to desired locations. Intuitively, the robot leg excavation location affects the avalanche area, and an obstacle's relative position to the excavation affects how much

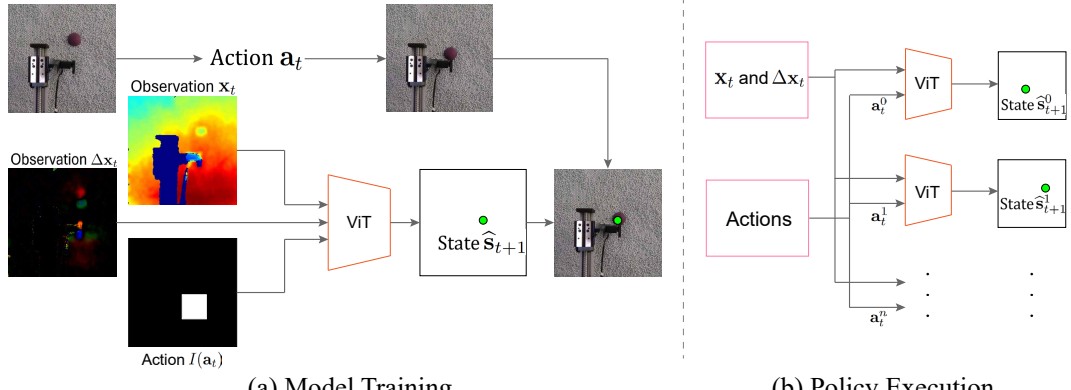

| (a) Model Training | (b) Policy Execution |

Figure 2: Overview of GRAIN. (a) The ViT has three inputs: the depth image of the granular surface $\mathbf{x}_t$, the change in depth before and after excavation $\Delta\mathbf{x}_t$, and the image representation of the action $I(\mathbf{a}_t)$. The output of the ViT is the prediction of the obstacle's post-excavation location which is a 2D vector. (b) We use the trained ViT to predict object movements based on leg excavation actions. The ViT combines the current observation with candidate actions as input and predicts corresponding object movement. The ViT considers one obstacle in its output; see Sec. 4.3 for handling $\geq 2$ obstacles at test time. See Sec. 4 for details on notation.

it is influenced by the avalanche. Due to the complexities of modeling avalanche dynamics and obstacle movements on the granular surface, we learn an action-conditioned dynamics machine learning model to predict obstacle movement. Similar learned models have predicted complex physical interactions for planning robot manipulation [61, 62, 63], suggesting their utility for our task. To represent the obstacle and the granular slope surface, we use a top-view depth image, $\mathbf{x}_t$.

Furthermore, during preliminary experiments, we observed that successfully relocating obstacles often required consecutive excavations at the same location. See Figure 3 for a visualization of grain flows with sequential excavations. To highlight this physical finding in our model, we use the change in the depth images of the granular slope surface between sequential excavation actions, denoted as $\Delta\mathbf{x}_t = \mathbf{x}_t - \mathbf{x}_{t-1}$ for $t > 0$, with $\Delta\mathbf{x}_t = 0$ (all pixels zero) if $t = 0$. We also introduce an image representation of robot leg excavation locations, which enables the action input to be spatially aligned with $\mathbf{x}_t$ and

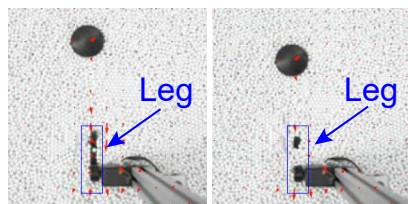

Figure 3: Granular flow for 2 sequential excavations. Red vectors represent the change of particle positions between two frames.

$\Delta\mathbf{x}_t$. This may help the network better learn the avalanche behavior compared to if the action is processed separately and concatenated with downstream visual features. We discretize excavation locations to 15 locations in a $5 \times 3$ grid (see Figure 5). We plot a white square with a side length the same as the robot leg length on a black background to represent the robot leg excavation location, $I(\mathbf{a}_t)$, where $\mathbf{a}_t$ is the excavation location on the 2D planar surface, and $I$ is a function that generates the RGB image representation of $\mathbf{a}_t$.

## 4.2 Training Objective for the Dynamics Model

We train a ViT to learn granular avalanche dynamics from C-shape robot leg excavations. The ViT predicts one obstacle movement on the granular slope, and we deal with multiple obstacles at test-time by repeatedly querying the model (see Sec. 4.3). The training loss $\mathcal{L}$ is defined as:

$$\mathcal{L}(\mathbf{x}_t, \Delta\mathbf{x}_t, \mathbf{a}_t, \mathbf{s}_{t+1}) = \|F_\theta(\mathbf{x}_t, \Delta\mathbf{x}_t, I(\mathbf{a}_t)) - \mathbf{s}_{t+1}\|_2, \tag{1}$$

where $F_\theta$ is the ViT parameterized by $\theta$ that takes, as input, a channel-wise concatenation of three spatially-aligned images: the depth image $\mathbf{x}_t$, the change in depth $\Delta\mathbf{x}_t$, and the action representation $I(\mathbf{a}_t)$. The ViT outputs the predicted post-excavation obstacle position $\widehat{\mathbf{s}}_{t+1} = F_\theta(\mathbf{x}_t, \Delta\mathbf{x}_t, I(\mathbf{a}_t))$, which we compare with the ground truth obstacle position $\mathbf{s}_{t+1}$ on the inclined 2D surface at time $t + 1$. Since the ViT predicts continuous values, we modify the ViT's default MLP classification header with an MLP regression header. See Figure 2 (left) for an overview of training.

### 4.3 Leg Manipulation Policy

We propose a greedy strategy for manipulation, which means the robot leg performs the excavation action at the location that has a maximum obstacle movement projection on the current line connecting the desired location and obstacle center toward the target area, described in Eq. 2:

$$\mathbf{a}_t^* = \arg \max_{\mathbf{a}_t \in \mathcal{A}} \; \mathbf{e}_t^T(F_\theta(\mathbf{x}_t, \Delta\mathbf{x}_t, I(\mathbf{a}_t)) - \mathbf{s}_t), \tag{2}$$

where $\mathcal{A}$ is the set of 15 possible actions (see Figure 5), and $\mathbf{e}_t^T$ is a 2D unit vector that points from the obstacle center to the target location at time $t$ (see Figure 4, target is the shaded red square). Parameters such as leg insertion depth and the angle of the excavation could also offer additional terrain-manipulation opportunities. As an initial step, we focus on demonstrating the effect of leg excavation *location*. Prior work has shown that a greedy policy for planning can be useful for object relocation [64], and we hypothesize the same may be true in our setting.

Figure 4: Example of $\mathbf{e}_t$ vector, pointing to the obstacle's target.

Our model is trained using robot excavation actions with a *single* obstacle. To use the model with *multiple* obstacles, we treat multiple obstacle movement prediction as a collection of independent predictions of a single obstacle movement. In particular, we randomly select an obstacle and mask out other obstacles on the depth image input, $\mathbf{x}_t$, where we use a window that has a length 3 times the obstacle radius to compute the average pixel values in the windows and replace the pixel value of the obstacle with the computed averaged pixel value. Furthermore, we mask out other obstacles in $\Delta\mathbf{x}_t$, where we set the pixel values to 0 corresponding to other obstacle positions. We repeat this process for all obstacles and get the predictions of all obstacle movements. We modify our manipulation policy to fit the task; the policy now considers the sum of obstacles projected movement on lines that connect their centers to their desired locations as described in Eq. 3:

$$\mathbf{a}_t^* = \arg \max_{\mathbf{a}_t \in \mathcal{A}} \sum_{\mathbf{e}_t^T, \mathbf{s}_t} \mathbf{e}_t^T(F_\theta(\widetilde{\mathbf{x}}_t, \widetilde{\Delta\mathbf{x}}_t, I(\mathbf{a}_t)) - \mathbf{s}_t), \tag{3}$$

where images $\widetilde{\mathbf{x}}_t$ and $\widetilde{\Delta\mathbf{x}}_t$ are masked versions of $\mathbf{x}_t$ and $\Delta\mathbf{x}_t$.

## 5 Experiment Setup

Existing simulators used in learning-based legged robot manipulation research, such as PyBullet [65], MuJoCo [66], or IsaacGym [67], do not support realistic robot interaction on granular media surfaces. Thus, we do all data collection, training, and experiments directly in the real world.

**Experiment environment:** Figure 1 illustrates our experiment setup. The main structure of the testbed is a granular trackway ($60\,\mathrm{cm}\ L \times 60\,\mathrm{cm}\ W \times 20\,\mathrm{cm}\ D$) filled with 6 mm plastic BBs (Matrix Tactical Systems). The 6 mm particles were chosen as they behave rheologically similar with natural sand and soil [68, 1, 69], while the simpler geometry and larger size facilitate image-based model training. The granular trackway can be tilted up to 35 degrees to emulate a wide variety of sand slopes [11] in natural environments. To study the avalanche dynamics and object movement upon different leg excavation actions, we build a gantry system with two linear actuators (one moves along the $x$ axis and another along the $y$ axis) to move a C-shape robot leg on a 2D surface above the granular slope. The C-shape robot leg has a diameter of 6.0 cm and a width of 2.0 cm, and the rotation center of the leg is 1.0 cm above the initial granular slope surface. The rotation frequency of the robot leg is fixed at 0.33 Hz. This is sufficiently above the granular surface, so the robot avoids touching it while transitioning between consecutive excavation actions. We mount an RGBD camera (Intel RealSense 435-i) above the granular slope to record the granular flow and obstacle movement.

**Data collection:** We collect a dataset of 100 trials, where each trial has 10 excavation actions. The time interval between two consecutive excavation actions is 12 s to enable the robot leg to transit to different locations. Before each trial, a human operator manually smoothed the granular media to a

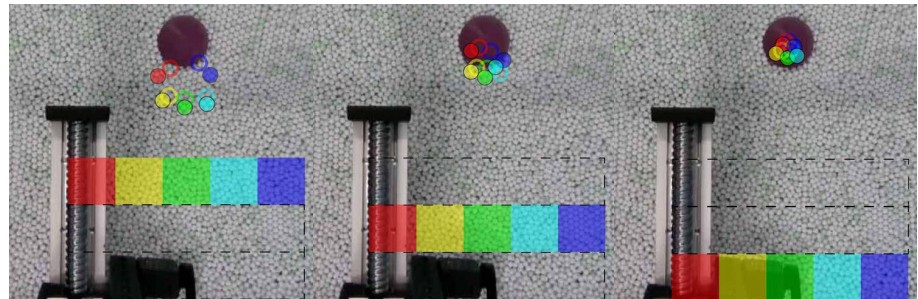

Figure 5: Obstacle movement with different excavation actions, visualized with colored squares (5 actions per image above). These $5 \times 3 = 15$ colored squares are the 15 discretized excavation actions we consider. The corresponding *solid* color circles and *empty* color circles are the trained model's predicted obstacle positions and experiment-measured obstacle positions, respectively. (This figure is best viewed zoomed-in.)

(roughly) even granular slope with an inclination angle $\Phi = 18$ degrees. This inclination angle is close to the angle of repose [70] of the granular material used in our work, which facilitates the study of the avalanche dynamics. Once the granular surface is prepared, the human placed a 3D-printed (PLA) obstacle on the granular surface at different locations relative to the leg. For all 100 trials, a semi-spherical obstacle with a 5 cm diameter is used. The RGBD camera has a video streaming rate of 15 Hz, and it collects 640x480 RGBD images after every excavation action. The ground truth obstacle movement is calculated based on the post-excavation RGBD image. Among the 100 trials, 36 use the same excavation location but different initial obstacle positions, 30 use the same initial obstacle position but different excavation locations, and 34 vary both the initial obstacle positions and excavation locations for each action. We use this data to train $F_\theta$.

**Evaluation:** For each trial, after the human places the obstacles, a computer program randomly selects target location(s) for each obstacle. Targets can be anywhere in the robot excavation action space, as long as they do not overlap with each other. Then, the program randomly selects a method among a set of methods we test (GRAIN, a baseline, or an ablation, see Sec. 6.2) for manipulation, to reduce human bias in making initial settings easier for our method. We evaluate our dynamics model and manipulation outcomes based on the prediction error for each excavation action, using Mean Absolute Error (MAE). In test trials, we compute performance based on the final distance between the obstacle center and its desired location. The success threshold is below 2.5 cm (the radius of the obstacle) which we measure via converting pixel distances in images to centimeters. We also use the average error between prediction and ground truth as a performance evaluation.

## 6 Experiment Results and Discussions

### 6.1 Model Performance on Predicting Obstacle Movements

To evaluate our model's performance in predicting obstacle movements, we place an obstacle on an undisturbed granular slope and test 15 excavation action locations; we reset the obstacle to the same spot after each action. See Figure 5 for a comparison between the predictions and the ground truth locations. The MAE for these 15 excavation actions is 1.13 cm, below our 2.5 cm threshold. Based on the promising MAE results, we use our trained model for planning in Sec. 6.2 and Sec. 6.3.

### 6.2 Single Leg Manipulation Performance

We evaluate GRAIN in real-world experiments with a single leg using four types of tasks:

1. **Single obstacle with a single task:** Using $K = 1$ obstacle, with a target position $\mathbf{p}$.
2. **Single obstacle with sequential tasks:** Using $K = 1$ but with an additional (second) target $\mathbf{q}$.
3. **Multiple obstacles:** Using $K = 4$ obstacles with targets $\{\mathbf{p}^{(1)}, \mathbf{p}^{(2)}, \mathbf{p}^{(3)}, \mathbf{p}^{(4)}\}$.
4. **Unseen obstacle:** Using $K = 1$ obstacle with a target position $\mathbf{p}$, but where the new obstacle has a star shape and weighs twice as much as the standard obstacle we use.

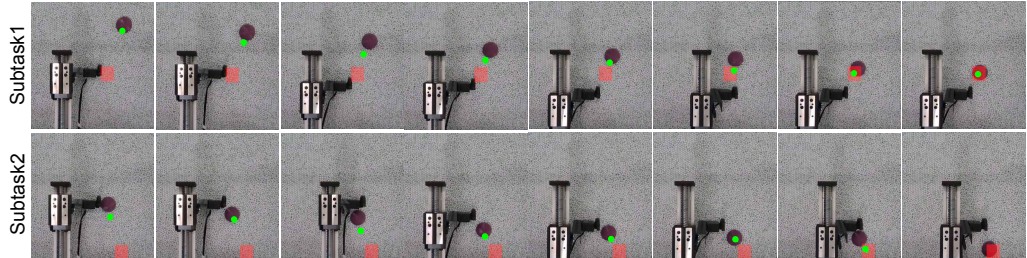

Figure 6: Single leg manipulation results, showing one trial of "Single obstacle with sequential tasks". The red shaded boxes are the obstacle targets. The green dots are the predicted post-excavation locations of obstacles after the leg performs an excavation at its location.

We compare GRAIN against a *baseline* and an *ablation*. Our baseline is an algorithmic, non-learning manipulation strategy. The baseline randomly selects an obstacle, and the robot performs the excavation at the closest available location to the target until the obstacle has met one of two termination criteria (see next paragraph). Then, this baseline randomly selects one of the remaining obstacles to manipulate and repeats until all obstacles are selected. Our ablation investigates the importance of our image representation of the robot excavation action to the successful training of our model. We train a ViT with a lower-dimensional, 2D *vector representation* of the robot excavation action. This cannot be directly channel-wise combined with the depth-based image observations, so we instead input it to the regression header of our ViT.

A manipulation trial terminates upon either of these conditions: (i) none of the excavation actions can take obstacles closer to their targets, or (ii) all obstacles have a small accumulated movement ($\leq$ 0.5 cm) over 3 sequential excavations. In (i), for the baseline method, we terminate when we observe that the previous excavation action took the obstacle away from its target or reached it.

Results from Tab. 1 suggest that GRAIN outperforms the baseline and the vector representation ablation. GRAIN has a success rate $\geq 80\%$ on all manipulation tasks, while the baseline has trouble with the "Multiple Obstacles" task, and the ablation has trouble with the "Unseen obstacle" task. The baseline particularly struggles when manipulating multiple obstacles (20% success), as it does not consider other obstacle movements when manipulating one obstacle. The ablation's granular dynamics model has a higher (i.e., worse) MAE in the predictions, which results in lower success rates versus GRAIN. We show one trial each of "Single obstacle with sequential tasks" and "Multiple obstacles" in Figure 6. We refer the reader to the supplementary website for videos.

| Task | Method | **MAE**$^{\dagger}$ (cm) | **MAE**$^{\S}$ (cm) | **Success rate** (%) |
|---|---|---|---|---|
| Single obstacle with single task | Baseline | N/A | 1.78($\pm$ 0.96) | 80% |
| | Vector representation | 2.34($\pm$ 0.94) | 1.76($\pm$ 0.63) | 80% |
| | GRAIN | 1.66($\pm$ 0.62) | 1.39($\pm$ 0.41) | 100% |
| Single obstacle with sequential tasks | Baseline | N/A | 2.24($\pm$ 1.12) | 60% |
| | Vector representation | 2.51($\pm$ 0.96) | 2.17($\pm$ 0.72) | 60% |
| | GRAIN | 1.78($\pm$ 0.62) | 1.42($\pm$ 0.45) | 100% |
| Multiple obstacles ($K = 4$) | Baseline | N/A | 5.12($\pm$ 3.44) | 20% |
| | Vector representation | 2.89($\pm$ 1.11) | 2.21($\pm$ 0.88) | 60% |
| | GRAIN | 2.21($\pm$ 0.91) | 1.85($\pm$ 0.74) | 80% |
| Unseen obstacle | Baseline | N/A | 4.34($\pm$ 2.87) | 60% |
| | Vector representation | 2.67($\pm$ 1.42) | 2.49($\pm$ 1.38) | 40% |
| | GRAIN | 1.96($\pm$ 0.90) | 1.91($\pm$ 0.73) | 80% |

$^{\dagger}$MAE for model predictions versus ground truth. $^{\S}$MAE for final obstacle positions versus target positions.

Table 1: We compare the quantitative performance of our method (GRAIN) versus alternative methods on four manipulation tasks. We report MAE in two columns, both using the format: Mean($\pm$ Standard Deviation), over 5 trials each. The "Baseline" method involves no prediction, hence the "N/A" in the first MAE column.

## 6.3 Quadruped Robot Manipulation Experiments

We evaluate GRAIN on the quadrupedal robot with 2 front legs as manipulators; we do not use the back legs since the robot body would likely block obstacles. Specifically, we use the same trained model as in Sec. 6.2 but we query the model at the 2 excavations where the legs are located, instead of the full set of 15. We show one trial in Figure 7. Results in Tab. 2 suggest that GRAIN achieves higher performance (i.e., lower MAE) than the baseline or vector representation methods.

| Task | Method | $\text{MAE}^{\dagger}$ (cm) | $\text{MAE}^{\S}$ (cm) | Success rate (%) |
|---|---|---|---|---|
| Multiple obstacles ($K = 3$) | Baseline | N/A | 4.56($\pm$ 2.46) | 30% |
| | Vector representation | 3.19($\pm$ 1.34) | 2.67($\pm$ 1.12) | 40% |
| | GRAIN | 2.52($\pm$ 1.08) | 1.91($\pm$ 0.97) | 70% |

$^{\dagger}$MAE for model predictions versus ground truth. $^{\S}$MAE for final obstacle positions versus target positions.

Table 2: We compare the performance of GRAIN and alternative methods on the multiple obstacles task (with 3 obstacles) using the quadrupedal robot, in a similar format as Tab. 1, except statistics are over 10 trials each.

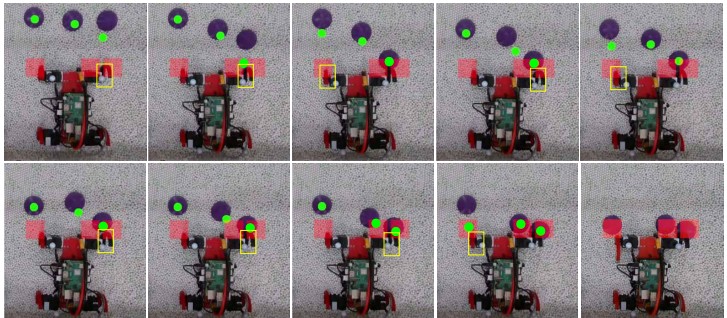

Figure 7: One trial of a quadruped robot manipulating three obstacles to reach targets (indicated with red shaded boxes). The yellow boxes highlight the leg that performs the excavation action at each step, and the green dots are the predicted post-excavation locations of obstacles after the leg performs the action.

## 6.4 Failure Cases and Limitations

A common failure of GRAIN with multiple obstacles occurs when the robot leg moves one obstacle to its target while simultaneously moving other obstacles too far from their targets. See Fig. 8 for an example, where two obstacles are already at their target locations (red shaded boxes), but the robot has trouble moving the third obstacle to its target (white arrow and green shaded box). The rightmost obstacle has reached its target, and further excavations will move it away from its target. Our manipulation policy only predicts obstacle movements for one step, and thus can have difficulty with tasks that require multiple-step planning, which we plan to address in future work.

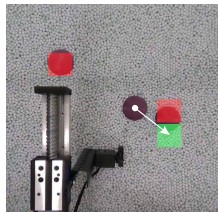

Figure 8: The arrow shows the robot cannot move the obstacle to its target (green square).

## 7 Conclusion

In this work, we present GRAIN, a method for a legged robot to indirectly manipulate obstacles on a granular surface. Our method trains a neural network to predict where an obstacle on a granular surface moves as the result of a legged robot's excavation action, enabling a quadrupedal robot to leverage granular avalanche dynamics to relocate obstacles. To our knowledge, this is the first work to explore indirect manipulation on deformable surfaces via legged robots. As the first step towards the complex problem of indirectly manipulating obstacles via strategic granular avalanches, we used granular particles of relatively simple geometry, and primarily focused on individual obstacle movement prediction with minimal inter-obstacle interactions. Prior work has shown that these simpler geometry particles behavior qualitatively similar in terms of rheology and interaction force laws [1], suggesting strong potential for GRAIN to be extended to more complex scenarios in future work.

**Acknowledgments**

This work is supported by funding from the National Science Foundation (NSF) CAREER award #2240075, the NASA Planetary Science and Technology Through Analog Research (PSTAR) program, Award # 80NSSC22K1313, and the NASA Lunar Surface Technology Research (LuSTR) program, Award # 80NSSC24K0127. The authors would like to thank Luke Cortez for helping with preliminary data collection, and Vedant Raval for helpful writing feedback.

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

# A   Additional Details of GRAIN

## A.1   Algorithm for Image Representation of Robot Excavation Action

To highlight the relationship among the robot excavation action, avalanche behavior and obstacle movement, we introduce our image representation of the excavation actions in Sec. 4.1. The pseudocode in Alg. 1 shows how we obtain this representation. We show several examples of the image representation in Figure 9, for different locations of the robot leg.

---

**Algorithm 1** Image Representation of Robot Excavation Action

---

**Require:** White square size $A$, coordinates of robot excavation action location $\mathbf{a}_t = (x_t, y_t)$
**Ensure:** Output image $O$ of size $(H, W)$
1: $O \leftarrow$ Zero matrix of size $(H, W)$
2: $x_{\text{start}} \leftarrow \max(0, x_t - \frac{A}{2})$
3: $y_{\text{start}} \leftarrow \max(0, y_t - \frac{A}{2})$
4: $x_{\text{end}} \leftarrow \min(H, x_t + \frac{A}{2})$
5: $y_{\text{end}} \leftarrow \min(W, y_t + \frac{A}{2})$
6: **for** $i = x_{\text{start}}$ to $x_{\text{end}}$ **do**
7:     **for** $j = y_{\text{start}}$ to $y_{\text{end}}$ **do**
8:         $O[i, j] \leftarrow 255$
9:     **end for**
10: **end for**

---

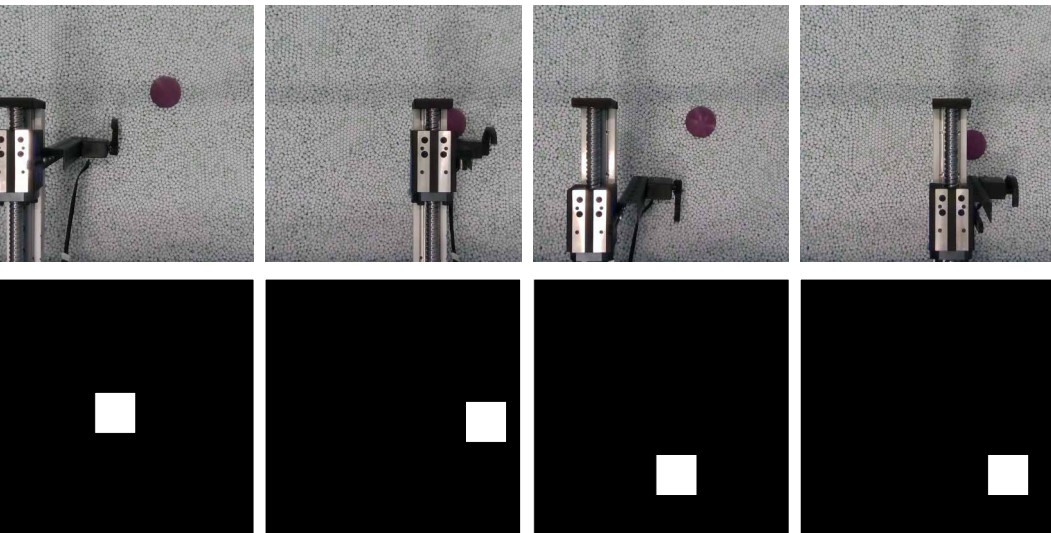

Figure 9: Additional examples of image representations of excavation actions. **Top**: RGB images of robot excavation actions. **Bottom**: The corresponding image representations of robot excavation actions.

## A.2   Algorithm for Masking Obstacles in $\mathbf{x}_t$

Sec. 4.3 discusses how we handle multiple obstacles. In Alg. 2, we formalize our method to mask other unselected obstacles. This lets the masked images look similar to images from the single obstacle case, which we use for training the ViT. See Figure 10 for an example of an image and its corresponding masked version (obtained via Alg. 2). During training, we use `cv2.colormap`, a package in `opencv`, to convert depth to RGB, as shown in the figure. For visual clarity, we overlay "Obstacles" and "Masked Obstacles."

---

**Algorithm 2** Masking Unselected Obstacles in Multiple Obstacles Manipulation

---

**Require:** Input image $I$ of size $(H, W)$, window size $B$, list of obstacle center coordinates $\{(x_k, y_k)\}_{k=1}^{K-1}$, list of pixels sets (obstacles) $\{P_k\}_{k=1}^{K-1}$ for each coordinate
**Ensure:** Output image $O$ of size $(H, W)$
1: $O \leftarrow I$
2: **for** each $(x, y) \in \{(x_k, y_k)\}_{k=1}^{K-1}$ with corresponding pixels set $P$ **do**
3:      $x_{\text{start}} \leftarrow \max(0, x - \frac{B}{2})$
4:      $y_{\text{start}} \leftarrow \max(0, y - \frac{B}{2})$
5:      $x_{\text{end}} \leftarrow \min(H, x + \frac{B}{2})$
6:      $y_{\text{end}} \leftarrow \min(W, y + \frac{B}{2})$
7:      $S \leftarrow \{I[i, j] \mid x_{\text{start}} \leq i < x_{\text{end}}, y_{\text{start}} \leq j < y_{\text{end}}\}$
8:      $\text{avg} \leftarrow \frac{1}{(x_{\text{end}} - x_{\text{start}})(y_{\text{end}} - y_{\text{start}})} \sum_{(i,j) \in S} I[i, j]$
9:      **for** each $(i, j) \in P$ **do**
10:         **if** $x_{\text{start}} \leq i < x_{\text{end}}$ and $y_{\text{start}} \leq j < y_{\text{end}}$ **then**
11:            $O[i, j] \leftarrow \text{avg}$
12:         **end if**
13:      **end for**
14: **end for**

---

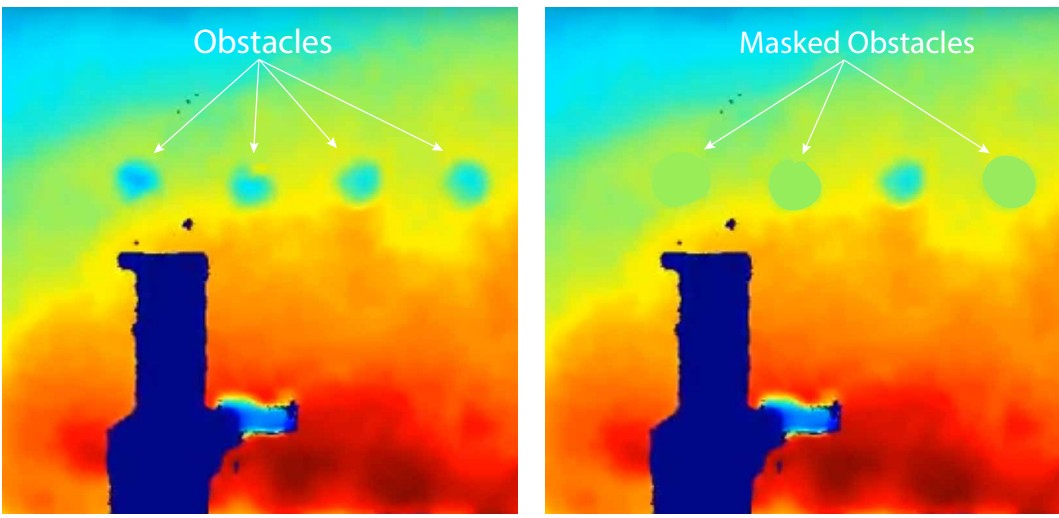

Figure 10: Example of masked image $\widetilde{\mathbf{x}}_t$. The left image is an example of $\mathbf{x}_t$ and the right image is the corresponding masked image $\widetilde{\mathbf{x}}_t$. Specifically, the second to the right most obstacle is the selected obstacle and other 3 obstacles are masked.

## A.3 Neural Network and Hyperparameter Details

Our overall data includes 10,437 images, which we obtain from 100 experiment trials. We split this data into 70% for training, 15% for validation, and the last 15% for testing. The input to our ViT is an image representation stacked channel-wise into a $7 \times 128 \times 128$ input. Our ViT has 46M trainable parameters, and is a smaller version of the ViT-Base model [15] so that we could fit it on our GPU. We train the ViT from scratch. To apply ViT to our task, we change the last layer of the ViT so that it has an MLP header that outputs a 2D vector corresponding to the predicted obstacle center position on the granular slope. Moreover, as we have a continuous prediction instead of a discrete prediction, we used MAE as the loss function instead of cross-entropy loss. We include our hyperparameters in Table 3.

| Hyperparameter | GRAIN Value |
|---|---|
| learning rate | 5e-6 |
| weight decay | 1e-3 |
| # encoder layers | 6 |
| # MLP layers | 2 |
| # heads | 6 |
| feedforward dimension | 768 |
| hidden dimension | 3072 |
| embedding dimension | 768 |
| input size | 7x128x128 |
| patch size | 16 |
| dropout | 0.1 |
| attention dropout probability | 0.1 |

Table 3: Hyperparameters for the Vision Transformer (ViT) we used in GRAIN.

# B  Additional Experiment Details

## B.1  Single Leg Manipulation Results

Figure 11 shows one experiment trial for three manipulation tasks: "Multiple obstacles ($K = 4$)," "Single obstacle with single task," and "Unseen obstacle." We refer the reader to Section 6.2 for a description of what the tasks mean, and for example trials from other tasks. The statistics of all manipulation trials are shown in Tab. 1.

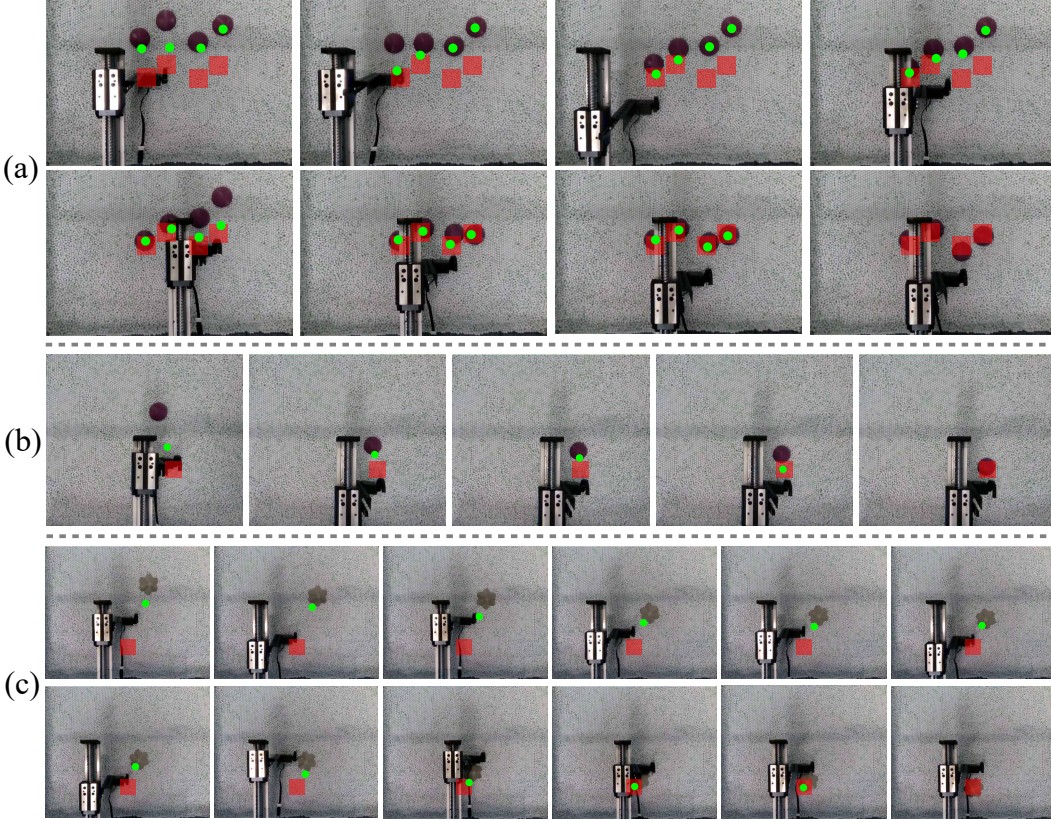

Figure 11: Single leg manipulation experiment results: (a) is Multiple obstacles ($K = 4$), (b) is "Single obstacle with single tasks," and (c) is "Unseen obstacle." In the above, red shaded areas are target areas. The green dots are the prediction of post-excavation locations of obstacles after the leg performs the action.

## B.2 Multiple Unseen Obstacles Manipulation

To test our trained model's generalization ability, we use obstacles with different shapes and weights in this task. Specifically, we use a star shape obstacle, a cuboid obstacle that has half of the weight of the obstacles used in the training dataset, and a hemisphere obstacle the same size but 4 times the weight of the obstacles used in the training dataset. All obstacles are 3D-printed. We place these unseen obstacles on the granular slope plus the obstacle we use in the training dataset as a total of $K = 4$ obstacles with a random distribution and executed the manipulation policy. We show one manipulation trial in Figure 12. Our system succeeds in 3 out of 5 trials.

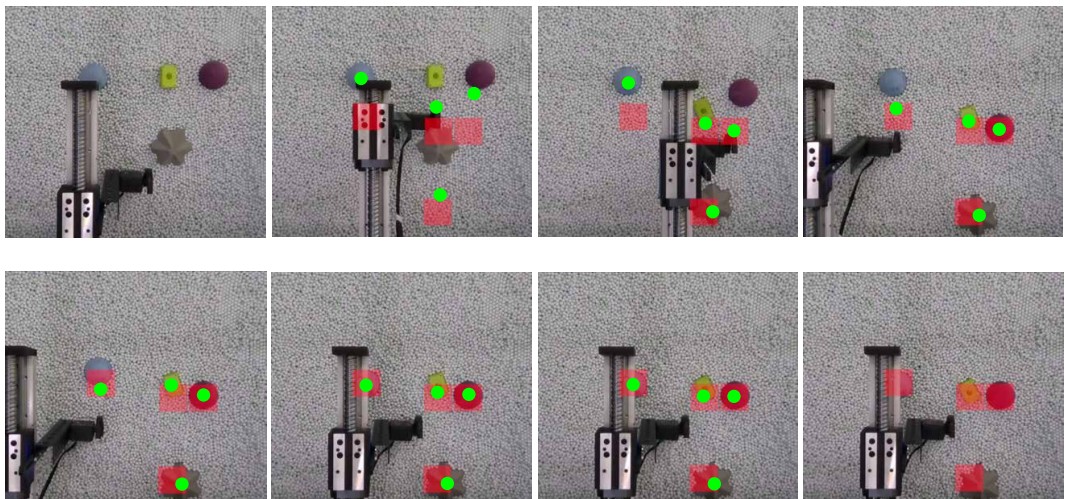

Figure 12: Multiple Unseen Obstacles Manipulation. Red shaded areas are target areas. The green dots are the prediction of post-excavation locations of obstacles after the leg performs an action at its current location.

