# OpenReview forum: "Learning Granular Media Avalanche Behavior for Indirectly Manipulating Obstacles on a Granular Slope"
_robot-learning.org/CoRL/2024/Conference — CoRL 2024_

### Official Review · Reviewer_M9RF · 2024-07-17
**Review of Submission153**

**Originality:** 3
**Technical Quality:** 4
**Clarity Of Presentation:** 4
**Potential Impact:** 3
**Recommendation:** 4
**Confidence:** 5

**Review:**

The paper investigates a challenging aspect of robotics related to granular physics, which is notoriously difficult to model accurately. Due to the complex and unpredictable nature of granular media, traditional simulation environments are limited in their effectiveness. The authors address this hard-to-model problem by employing machine-learning approaches and conducting systematic experiments.

The paper is well-written and provides valuable insights that can contribute significantly to robotics. Specifically, this work has the potential to enhance robotic actuation principles in highly flowable and deformable terrains, such as those found in extraterrestrial environments.

Strengths:
By leveraging machine learning to predict and manipulate object movements on granular slopes, this research might offers solutions that could improve the performance and adaptability of legged robots operating in challenging conditions.

Weakness:
The granular terrain used in the study consists of spherical plastic particles, which do not naturally occur in the real world. As a result, even though the insights gained are valuable, the findings may not be directly transferable to real-life robotic scenarios. Additionally, the authors could have leveraged a simulation environment like Chrono, which facilitates the simulation of granular particles and multi-body robots, for more extensive learning and experimentation

**Quality Of The Limitations Section:**

3

**Questions For Rebuttal:**

1) In Abstract: Relocate the statement  starting with "Supplementary material ..."

2) I recommend adding the following papers to the Related Work section, as they are pertinent to learning granular terrain manipulation for locomotion purposes.

    Real‐time remodeling of granular terrain for robot locomotion

    Learning manipulation of steep granular slopes for fast Mini Rover turning

3) I suggest providing more explanation in the text as to why the authors chose 6 mm plastic particles.

4) I wonder if the movement of one leg affects multiple objects at the same time during manipulation (Fig 7)

5) I suggest enhancing (and enlarging the related region) Fig 5 for clarity.

6) The conclusion section does not seem to reflect the importance of the findings of the paper.

**Robotics Focus:**

4

**Summary Of Paper:**

Inspired by living organisms this work uses avalanche dynamics for indirect object manipulation on granular slopes. This paper employs a Vision Transformer (ViT) to process images of granular dynamics and robot actions. The ViT predicts object movement, guiding the robot's leg excavation actions. Trained with data from various trials, the model performs well in new and unseen settings, achieving over 80% success in various tasks with up to four obstacles and generalizing to objects with different physical properties.

**Summary Of Recommendation:**

I recommend accepting the paper as it presents a novel approach to robotic manipulation of granular surfaces for displacing objects, which could potentially be used for obstacle-aided locomotion purposes.

---

### Official Review · Reviewer_egEL · 2024-07-20
**Learning Granular Media Avalanche Behavior for Indirectly Manipulating Obstacles on a Granular Slope**

**Originality:** 3
**Technical Quality:** 3
**Clarity Of Presentation:** 3
**Potential Impact:** 2
**Recommendation:** 3
**Confidence:** 3

**Review:**

The research presented a visual-transformer based method for obtaining image representations of granular materials to improve robot excavation (formulated as a quadruped with a ‘manipulation action’). The presented work outlines an interesting contribution towards robotic systems handling non-homogeneous granular materials, which as the author states, given the lack of realistic support on simulators, is an important challenge to tackle. Hence, using a visual transformer combined with an RL agent is a novel way of tackling the problem. The quality of the proposed method is evidenced when compared with other methods and demonstrates a successful outcome with respect to task completion. In addition, the contributions outlined seem to be supported by quantitative experiments. There clarity of the methods could be improved by adding more detail on the implementation and hyperparameters, including github repository/documentation for replicating the experiments. Overall, the work presents a significant contribution towards using robots with granular materials, where the main aspect is that vision transformers seem to help with the task. It would be great to outline what is the computational cost for achieving this. The main strength of this work is that they carried out their method successfully on the robotic platform, whereas the main weakness is that the baselines/ablation chosen are quite simple so it does not fully place this method with other related work e.g. CNN-based or other related works in manipulating granular materials.

**Quality Of The Limitations Section:**

2

**Questions For Rebuttal:**

* In your related work you mention: ‘In some of the most relevant prior work, Schenck et al. [36] use an image-based representation of granular media and train a convolutional neural network to predict changes in the granular media state.’ Could you elaborate more on why you decided to use a vision transformer rather than a CNN-based method? Did you try to use a CNN-based method and deemed it insufficient? It would be nice to add these results to your experimental evaluation.
* How representative is the obstacle of real-world obstacles in this domain? From looking at the appendix, it seems that you have used very simplified methods for dealing with obstacles (opencv-based methods). Was there a reason why you did not explore other segmentation methods e.g. segment anything model that would let you generalise to more complex obstacles?
* Did you explore your method with other terrains e.g. soil? How would it perform with a material that might be more challenging than sand? It would be great if you can add experiments/comments on this.
* ‘Prior work has shown that a greedy policy for planning can be useful for object relocation tasks [60], and we hypothesize that the same may be true for our setting.’ - how did you conclude this? What other policy strategies did you consider and do you have any quantitative results for these?
* ‘Since the ViT predicts continuous values, we modify the ViT’s default MLP classification header with an MLP regression header.’ - this is slightly confusing - did you just change the loss function here?
* ‘We train a ViT to learn granular avalanche dynamics from C-shape robot leg excavations.’ - how big is your training data? Did you train from scratch or fine-tune? Could you elaborate more on how this network looked like, what was the training/evaluation performance of this model?

Minor comments:
- Refer to images as Figure rather than Fig in the main text.
- Hyperparameters for your models are not tabulated anywhere. Please add to the appendix to make the work more reproducible.
- Figure 2 should have the mathematical notation in the arrows and the text in the caption.
- Figure 4 and Figure 8 captions need to be expanded.
- The text wrapping around the figures is a bit confusing. I would advise the authors to rethink the layout of the paper.
- The conclusion is a bit short - maybe remove the failure case image so you have more space for the conclusion?

**Robotics Focus:**

4

**Summary Of Paper:**

In their paper, the authors propose their method (Granular Robotic Avalanche Interaction - GRAIN) for using vision transformers to obtain image representations of granular dynamics and robot excavation actions. They collect data from 100 real physical trials & their results show that their model has a success rate of 80%+ to predict object movements with up to 4 obstacles. They summarise their main contributions as the following: In summary, our main contributions are as follows: (1) novel problem formulation of using legged robots and leg excavation actions to indirectly manipulate obstacles to targets over complex, granular terrains; (2) their GRAIN method - a novel approach that uses image-based representations of granular dynamics and legged excavation actions to predict object movements on a granular surface; (3) experimental results showing that a legged robot using GRAIN can move obstacles to targets, and that GRAIN outperforms an ablation and a non-learning baseline.

**Summary Of Recommendation:**

The paper could be interesting to the CORL community, should the authors address the questions presented in the rebuttal section. Given the efforts the authors put into carrying it out onto a real robotic platform, I believe this is achievable and would strengthen their work for publication.

---

### Official Review · Reviewer_fr2L · 2024-07-21

**Originality:** 4
**Technical Quality:** 3
**Clarity Of Presentation:** 4
**Potential Impact:** 3
**Recommendation:** 2
**Confidence:** 4

**Review:**

Overall, I like the problem setup. Manipulating granular material that is subject to complex dynamics clearly benefits from a learning-based solution. I also like the data collection setup. It is quite clever to gather data with a fixed gantry. The paper is overall well-written.

My concerns listed in the order of importance to my rating:
1. The authors claim in Line 64-65 that none of the prior works on granular material manipulation is comparable because the problem setup in this paper aims to manipulate an object through the granular medium. I disagree with this claim. There are quite a few works that study generalized inter-object dynamics, such as [1-2]. The methods can be easily applied to the problem setup by adopting the action discretization and either changing the output to the obstacle location or directly reading out the object location if a particle-based such as [1] is used.  [2] also uses a similar image-based action representation as GRAIN. I don’t think changing the output format can exempt the proposed method from comparison with closely-related prior research.
- [1] Dynamic-Resolution Model Learning for Object Pile Manipulation Yixuan Wang, Yunzhu Li, Katherine Driggs-Campbell, Li Fei-Fei, Jiajun Wu, RSS 2023
- [2] Neural Field Dynamics Model for Granular Object Piles Manipulation, Shangjie Xue, Shuo Cheng, Pujith Kachana, Danfei Xu CoRL 2023

2. Discretizing excavation action to 15 bins seems to be a very strong assumption to make. I would assume for such an intricate task, more fine-grained action representation can help the model choose a more optimal solution. Moreover, I would expect the magnitude and the angle of the excavation action would affect the outcome significantly. I would suggest at least comparing with different discretization resolutions.

3. Merging independent predictions about each obstacle to approximate prediction for multi-obstacle scenarios seems to be a significant simplification. I would assume there are inter-obstacle interactions and obstacle-granular material interaction that gets affected by the relative configurations of the obstacles. The paper does not offer any justification why such a simplification is ok.

4. How does the quadruped robot execute sequential actions at different locations in one single trial? Does it move to a new excavation location on its own? If so, it seems that the action for locomotion would affect the granular material.

5. I would greatly appreciate a video demonstration of the system, as the system performance is best understood with animated motion.

**Quality Of The Limitations Section:**

2

**Questions For Rebuttal:**

See "weakness" section

**Robotics Focus:**

4

**Summary Of Paper:**

The paper proposes a method to manipulate obstacles on a granular slope (surface covered in granular material such as sand). The key idea is to cause avalanche behavior of the granular material using the robot’s end effector, in this case, the leg of a quadruped robot. This behavior indirectly moves the obstacle to a desired location such that the robot can find a collision-free path. The technical method is an action-condition model that predicts future obstacle movement subject to such maneuver. Both data collection and evaluation is done in a real-world environment. The data collected by repeatedly perturbing the granular slope with a leg-shaped manipulator mounted on a fixed gantry. The trained model is also transferred to a quadruped robot. The model is evaluated based on the displacement between the desired end positions of the obstacles and the positions reached.

**Summary Of Recommendation:**

I would be willing to change my rating if the authors could adequately address my top 3 concerns.

---

### Author Rebuttal · Authors · 2024-08-13

We included our revised paper and rebuttal figures (for responses to reviewers) in the attached GRAIN(rebuttal).zip file.

---

### Decision · Program_Chairs · 2024-09-04

**Decision:**

Accept

**Comment:**

Summarizing the strengths and weaknesses pointed out by the reviewers:

Strengths:
- Well defined problem with complex dynamics that can benefit from a learning-based solution
- Visual transformer + RL is a novel solution to the problem
- Good data collection setup
- Results show the proposed method is effective
- Well-written

Weaknesses:
- Claims about prior work are incorrect (see reviewer's comments for details)
- The action space seems to discretized to be effective, but no exploration of this is done
- The object independence assumption is not well justified
- The paper does not state how the robot moves between action locations in a single trial
- Baseline/ablation is too simple to be an adequate comparison
- The granular media used (spherical plastic particles) has limited applicability to real world scenarios

Overall, the reviewers are split (1 weak reject vs. 1 weak accept + 1 strong accept). The concerns of the weak reject reviewer appear to be mostly addressable by adding content to the paper and running a small experiment. The authors should address as many concerns of the reviewers as they can.

**Update after rebuttal**: The authors' provided a rebuttal where they directly responded to the reviewers' comments and made minor modifications to their paper. None of the reviewers replied or updated their recommendations, so let us take a look and see if the rebuttal sufficiently addressed their concerns. 2 of the reviewers already recommended acceptance, so we will focus on reviewer fr2L, who recommended weak reject. The reviewer very helpfully numbered their concerns, so we will analyze each in order:

1. The authors' response was quite extensive. It appears as though they attempted to implement both the papers referenced by the reviewer. However, they were only able to partially implement 1 of them. They included a results figure in their rebuttal, but it is not in the paper. It would have been nice to include that in the paper/appendix. The metric used in this figure is training loss, not a great metric but okay (assuming it's loss on a hold-out test set). I would say, given these results aren't in the paper, this concern is only partially addressed.

2. Again, the authors added a figure in the rebuttal but not in the paper/appendix. They show a comparison of a finer discretization than they used in the paper, and the results are very similar, justifying their choice of 5cm bins. However, they only evaluate on a single row of actions, rather than the whole space. It would have been nice to see this on the entire experiment space.

3. No new results for this one, just the authors' acknowledging the limitations of assuming independence between the objects and saying that's for future work.

4. It be confusing from the paper to understand how these actions are done, but upon a close reading it is clear that the data is collected with a gantry robot, thus locomotion is not an issue.

5. The video link was updated, and the videos do help understanding the experimental setup better.

Overall, I'd say reviewer fr2L's concerns were only partially addressed. Nonetheless, given that the other 2 reviewers recommend acceptance, and reviewer M9RF strong accept, I would say this paper should be accepted.